# Development of Desiccation-Tolerant Probiotic Biofilms Inhibitory for Growth of Foodborne Pathogens on Stainless Steel Surfaces

**DOI:** 10.3390/foods11060831

**Published:** 2022-03-14

**Authors:** Jong-Hui Kim, Eun-Seon Lee, Kyoung-Ja Song, Bu-Min Kim, Jun-Sang Ham, Mi-Hwa Oh

**Affiliations:** National Institute of Animal Science, Rural Development Administration, Wanju-gun 55365, Korea; kimjh8199@korea.kr (J.-H.K.); les1023@korea.kr (E.-S.L.); ghkdgn219@korea.kr (K.-J.S.); scarlet7@korea.kr (B.-M.K.); hamjs@korea.kr (J.-S.H.)

**Keywords:** lactic acid bacteria, biofilm, stainless steel, desiccation

## Abstract

Lactic acid bacteria biofilms can be used to reduce foodborne pathogen contamination in the food industry. However, studies on growth inhibition of foodborne pathogens by inducing biofilm formation of antagonistic microorganisms on abiotic surfaces are rare. We developed a desiccation-tolerant antimicrobial probiotic biofilm. *Lactobacillus sakei* M129-1 and *Pediococcus pentosaceus* M132-2 isolated from fermented Korean foods were found to exhibit broad-spectrum antibacterial activity against *Bacillus cereus*, *Escherichia coli* O157:H7, *Staphylococcus aureus*, *Listeria monocytogenes*, and *Salmonella enterica*. Their biofilm levels were significantly (*p* < 0.05) higher on stainless steel than on polyethylene or ceramic. Biofilms of both isolates showed significantly (*p* < 0.05) enhanced resistance against desiccation (exposure to 43% atmospheric relative humidity) as compared with the isolates not in the biofilm form. The antimicrobial activity of the isolates was sustained in dried biofilms on stainless steel surface; the initial number of foodborne pathogens (average 7.0 log CFU/mL), inoculated on stainless steel chips containing *L. sakei* M129-1 or *P. pentosaceus* M132-2 biofilm decreased to less than 1.0 log CFU within 48 h. The lactic acid bacteria antibacterial biofilms developed in this study may be applied to desiccated environmental surfaces in food-related environments to improve microbiological food safety.

## 1. Introduction

Food contamination with pathogens is a serious public health problem that can cause foodborne diseases [1]. Annually, an estimated 600 million illnesses and 420,000 deaths are caused by foodborne diseases worldwide [2,3]. To mitigate this problem, hazard analysis and critical control point systems should be introduced in food production and processing plants along with stringent cleaning, disinfection, and sterilization protocols to ensure global food safety [4]. However, it may be difficult to maintain hygiene in areas of food processing equipment that come into contact with food but are out of reach.

Biofilms constitute aggregates of growing microorganisms attached to biotic or abiotic surfaces embedded in self-secreted extracellular polymeric substances (EPSs) [5]. These structures act as a barrier, protecting the microbial cells from environmental stresses, including drying, sanitizers, ultraviolet light, disinfectants, and antibiotic treatments [6]. *Listeria monocytogenes*, *Salmonella spp.*, *Bacillus cereus*, *Staphylococcus aureus*, and pathogenic *Escherichia coli* are the main causes of foodborne diseases and represent a concern, especially in the form of biofilms, because they can attach to virtually any type of food processing equipment surface, such as plastic, stainless steel, ceramic, and glass [7,8]. Biofilms thus represent a potential health risk as they grow despite stringent antibacterial conditions enforced in the food processing and preservation industry [9].

However, biofilms derived from Lactobacillus strains are generally recognized as safe and may provide a promising means of interfering with the formation of pathogenic biofilms due to their numerous beneficial properties [10,11]. For example, these beneficial biofilms can inhibit the growth of pathogenic and spoilage microorganisms [12]. Previous studies have reported that many antibacterial substances were secreted only when cells developed into biofilms or attached to surfaces with EPS production and that biofilms of lactic acid bacteria (LAB) exhibited higher productions of anti-inflammatory and antimicrobial compounds than liquid cultures [13,14]. In addition, the use of microorganisms in a direct manner has the advantage of maintaining antibacterial activity through continuous metabolism as compared with extracts, which may lose activity over time [15]. However, most of these studies have been conducted on planktonic-growing microorganisms, and the biological properties observed in suspended cultures may differ from those of cells that form biofilms. Moreover, growth inhibition of foodborne pathogens by inducing biofilm formation of antagonistic microorganisms on abiotic surfaces has received less research attention.

Therefore, in the current study, we focused on the biofilm formation ability of LAB as well as their antibacterial activity on pathogens present on abiotic surfaces of food processing equipment. The purpose of this study was to develop an antagonistic biofilm that is desiccation-tolerant and inhibits the growth of foodborne pathogens. Two LAB isolates from various fermented Korean foods were comparatively analyzed for their ability to form biofilms on polystyrene, stainless steel, and ceramic surfaces. Further, the desiccation resistance of the cells constituting the biofilm was evaluated, and finally, the antimicrobial activity of the biofilm against five foodborne pathogens on a stainless steel surface was confirmed.

## 2. Materials and Methods

### 2.1. Isolation and Identification of LAB

In total, 245 LAB strains were isolated from 111 *kimchi* samples (230 strains), 79 *doenjang* samples (33 strains), and 9 salted fish samples (9 strains). The samples were purchased from a local market (Jeonju, Korea) and each sample (10 g) was mixed with 90 mL of 0.1% peptone water and homogenized using a Stomacher 80 blender (Seward, Worthing, UK). Serially diluted fermented Korean food samples were spread on de Man, Rogosa and Sharpe agar (MRS) plates (BD Difco) and incubated at 37 °C for 3 days. To identify the isolates, purified polymerase chain reaction amplicons from the isolates were sequenced by Solgent (Daejeon, Korea) using universal primers 27F (5′-AGAGTTTGATCMTGGCTCAG-3′), 1492R (5′-TACGGYTACCTTGTTACGACTT-3′), and 1492R (5′-TACGGYTACCTTGTTACGACTT-3′). For identification, 16S rRNA sequences were searched using the BLAST program (http://www.ncbi.nlm.nih.gov/BLAST, accessed on 14 October 2021).

### 2.2. Screening of LAB Isolates for Antimicrobial Activity

The antimicrobial activity of LAB was evaluated using the well diffusion method with some modifications. Overnight bacterial cultures in MRS broth were centrifuged at 10,000× *g* for 7 min at 4 °C. The cell-free supernatants (CFS) were collected and filtered using 0.22-µm pore filter membranes (Millipore, Molsheim, France) to remove residual bacterial cells.

Five foodborne pathogenic bacteria (*Bacillus cereus* KCCM 40935, *Escherichia coli* O157:H7 ATCC 43888, *Listeria monocytogenes* CCARM 0214, *Staphylococcus aureus* ATCC 11778, and *Salmonella enterica* KCCM 12021) were grown in tryptic soy broth at 37 °C for 18 h [16] and diluted in physiological saline. Trypticase soy agar (TSA, BD Difco) plates were overlaid with 5 mL of TSA soft agar (0.75%) and inoculated with 100 µL of each pathogenic strain (7 log CFU/mL). A well of fixed diameter was created in the center of each plate and filled with CFS (80 µL). This was followed by incubation of the plates at 37 °C for 18 h. Antimicrobial activity was detected by measuring the zone of inhibition (including the well diameter) that appeared after the incubation period. Uninoculated MRS broth was used as a negative control. Each experiment was performed in triplicate.

### 2.3. Measurement of Cell Surface Hydrophobicity and Auto-Aggregation Ability of LAB

Bacterial adhesion of LAB to hydrocarbons was measured as previously described [17], with slight modifications. Briefly, LAB isolates cultured overnight were centrifuged at 8000× *g* for 5 min. The pellet was washed twice with sterile phosphate-buffered saline (PBS; pH 7.2) and resuspended in sterile PBS to an optical density of 0.5 (A_0_) at 600 nm wavelength. The bacterial suspension was then mixed vigorously with an equal amount of xylene (Sigma-Aldrich) and allowed to stand at room temperature for 1 h. The separated aqueous phase was carefully removed, and its absorbance at 600 nm was measured (A1). Surface hydrophobicity (H%) was calculated using the formula H% = (1 − A1/A0) × 100.

The auto-aggregation ability of the LAB strains was evaluated as described by Gómez et al. [12], with some modifications. Briefly, LAB isolates cultured overnight were centrifuged at 8000× *g* for 5 min. The pellet was washed twice with sterile PBS (pH 7.2) and resuspended in sterile PBS to an optical density of 0.5 (A0) at 600 nm wavelength. The absorbance at 600 nm of the homogenized bacterial suspension was again measured after the suspension had been standing undisturbed at 37 °C for 24 h (At). Auto-aggregation (A%) was calculated using the formula A% = (1 − At/A0) × 100. All experiments were performed in triplicate.

### 2.4. Biofilm Assay Using Crystal Violet Staining

Biofilm assays were performed colorimetrically using crystal violet staining as described by Merino et al. [18], with some modifications. Briefly, 2 mL of MRS broth was placed in each well of a 12-well plate inoculated with Lactobacillus strains (1%), and the plate was incubated at 37 °C for 72 h. Thereafter, the supernatant was discarded, and the wells were washed twice with PBS to remove non-adherent cells. This was followed by the addition of 1 mL of 0.2% crystal violet solution (in PBS) to each well and incubation at room temperature for 15 min, following which the dye was discarded and the wells washed three times with PBS. Thereafter, 600 μL of 96% ethanol was added to each well to extract the crystal violet. The extracted supernatant was transferred to a new well plate and absorbance at 595 nm wavelength was measured using a microplate reader (Biotek Synergy HT). To enumerate viable cells, supernatants were removed and biofilms were washed twice with PBS. Aliquots of 1 mL PBS were added to each well, and the cells were detached by scraping. Serial dilutions (1/10) of the bacterial suspension followed by plate counting on MRS agar were performed.

### 2.5. Biofilm Formation Analysis by Surface Type and Incubation Temperature and Duration

Biofilm analysis was performed as described by Kim et al. [19], with slight modifications. Three types of materials with widespread use in food processing environments were used: (1) polystyrene chips (PCs; 20 mm × 20 mm and 0.2 mm thickness); (2) stainless steel chips (SSCs; Type 403, 20 × 20 mm and 0.2 mm thickness); and (3) ceramic chips (CCs; 20 mm × 20 mm and 0.2 mm thickness). These were cleaned using sonication and sterilized at 120 °C for 15 min to eliminate grease and microbes. The chips were then placed in Petri dishes (90 mm diameter × 15 mm height; SPL, Seoul, South Korea) containing 20 mL MRS broth, then inoculated with 0.1% antagonistic bacterial culture and placed at 5, 15, 25, or 35 °C for 72 h to determine the effect of temperature. The chips, along with the attached cells, were then placed in a Petri dish and loosely attached cells were removed by gently rinsing the chips in 40 mL of PBS (pH 7.2). The biofilm mass formed on the chips was measured as previously described [20], with slight modifications. Shortly after, biofilm-attached chips were transferred to a 6-well plate, then 3 mL of crystal violet (0.2%) was added, followed by incubation at 25 °C for 15 min to stain the coupons. The excess dye on the coupons was subsequently washed off using deionized water, following which they were air-dried at 60 °C for 2 h. The dye bound to the cells was extracted using 850 μL of 96% ethanol. The concentration of the extracted dye is expressed as absorbance at 595 nm wavelength.

To determine the effect of incubation duration on biofilm formation, incubation was carried out at 25 °C for 1, 2, 3, or 6 days. The absorbance values of the biofilms attached to the chip were measured as previously described. All tests were performed in triplicate and three independent sets of tests were performed. MRS medium without inoculation was used as a negative control.

### 2.6. Analysis of Antimicrobial Activity of LAB Biofilms against Pathogens on SSCs

The antimicrobial activity of two antagonistic biofilms against five foodborne pathogens present on SSCs was studied as described by Kim et al. [19], with some modifications. Briefly, SSCs were immersed in 20 mL of MRS broth inoculated with two antagonistic bacterial suspensions (M129-1 and M132-2, 0.1%) and incubated at 25 °C for 72 h. Following the incubation period, the cell-attached chip was placed in a polystyrene dish and quickly rinsed with PBS, pH 7.2. After transferring the cell-attached chip to a new Petri dish, five phagocytic pathogen suspensions (100 μL; approximately 8 log CFU/mL) were inoculated on its surface (ca 8 log CFU/chip), as well as on sterile SSCs (controls). The inoculated SSCs were placed in a Petri dish and incubated for up to 48 h in an incubator with the internal atmosphere adjusted to 43% relative humidity (RH) and 25 °C. Following incubation for 0, 3, 6, 12, 24, or 48 h, the cell-attached chips were placed in a Petri dish and rinsed rapidly with PBS. Thereafter, the chips were thoroughly scraped with a scraper to separate the cells, which were then transferred to new tubes containing 30 mL of PBS and vortexed at maximum speed for 1 min. The PBS wash suspensions were serially diluted in saline solution, plated on TSA (Difco BD), and incubated at 37 °C for 24 h.

### 2.7. Analysis of Desiccation Tolerance of LAB Biofilms

Desiccation tolerance of LAB strains was studied as described by Kim et al. [19], with some modifications. Briefly, SSCs were placed in a Petri dish containing 20 mL MRS broth, inoculated with antagonistic bacterial culture, and incubated at 25 °C for 72 h to enable biofilm formation. SSCs containing biofilms were then placed in Petri dishes and incubated for up to 120 h in an incubator with the internal atmosphere adjusted to 43% RH and 25 °C. Following incubation for 0, 24, 48, 72, 96, or 120 h, the cell-attached chips were placed in a polystyrene dish and rinsed rapidly with PBS. Thereafter, the chips were thoroughly scraped with a scraper to separate the cells, which were then transferred to new tubes containing 10 mL of MRS broth and vortexed at maximum speed for 1 min. The MRS wash suspensions were serially diluted in peptone water, plated on MRS agar (Difco BD), and incubated at 37 °C for 24 h.

### 2.8. Statistical Analysis

Hydrophobicity, auto-aggregation, biofilm formation, and changes in the population of five pathogens with or without biofilm were obtained for each LAB isolate in triplicate (at least) and expressed as the mean ± standard deviation. One-way analysis of variance (ANOVA) and Duncan’s multiple range test were performed to determine the statistical significance between the treatments. Differences between means were considered significant at *p* < 0.05. All analyses were conducted using IBM SPSS statistics version 26 (SPSS Inc., Chicago, IL, USA).

## 3. Results

### 3.1. LAB Isolates Identified to Have Antibacterial Activity

A total of 245 LAB isolates were obtained from the fermented foods and screened for antibacterial activity against five foodborne pathogens: *B. cereus*, *E. coli* O157:H7, *L. monocytogenes*, *S. aureus*, and *Salmonella enterica*. As many as 51 isolates showed antibacterial activity against one or more tested pathogens; among these, five strains (M2-8, M102-3, M111-1, M113-5, and M129-1) isolated from *kimchi* samples and one strain (M132-2) isolated from *doenjang* samples showed high antibacterial activity against all five pathogens (Table 1). These six isolates, representing antagonistic bacteria with broad-spectrum antibacterial activity, were selected to evaluate their biofilm-forming ability. Using 16S rDNA sequencing, four isolates (M2-8, M111-1, M113-5, and M132-2) were identified as *Leuconostoc mesenteroides* (99.8% homology), *Leuconostoc lactis* (99.6% homology), *Lactobacillus curvatus* (99.9% homology), and *Pediococcus pentosaceus* (99.7% homology), respectively, whereas isolates M102-3 and M129-1 exhibited high sequence similarity (99.7 and 99.9% homology, respectively) with *Lactobacillus sakei*.

### 3.2. Surface Hydrophobicity, Auto-Aggregation, and Biofilm Formation of LAB Isolates

To examine the correlation between the bacterial surface properties and the capacity to form biofilms, the hydrophobicity, auto-aggregation, and biofilm formation values of the selected antagonistic LAB were analyzed. Bacterial adhesion to hydrocarbons is widely used to measure cell surface hydrophobicity; in this study, the hydrophobicity of the bacterial cell surface was measured by observing its adhesion to xylene (Table 2). Most of the strains were hydrophobic, especially M113-5, M129-1, and M132-2, which showed high hydrophobicity (adhesion to xylene between 68% and 72%). The auto-aggregation ability of the antagonistic LAB was also high (46–52% in M113-5, M129-1, and M132-2 strains) (*p* < 0.05). In crystal violet biofilm staining, M129-1 and M132-2 strains showed high biofilm-forming ability, but unexpectedly, M113-5, which had the highest hydrophobicity, showed an intermediate level of biofilm formation (*p* < 0.05). Thus, the strains M129-1 and M132-2 could have an advantage in antagonistic biofilm formation; therefore, these strains were selected for subsequent investigation of their biofilm properties.

### 3.3. Biofilm Formation by LAB Isolates on Different Surfaces and Under Different Incubation Temperatures and Durations

The results of biofilm formation by the two antagonistic isolates, M129-1 and M132-2, on different surfaces are shown in Figure 1. The ability of the two LAB strains to generate biofilms on the selected surfaces for 3 days at 4, 15, 25, and 35 °C is shown in Figure 1A. Biofilms were produced by M129-1 and M132-2 at significantly higher levels on SSCs at 5 °C (optical density [OD] values: 0.81 and 0.86, respectively), 15 °C (OD values: 1.23 and 1.35, respectively), and 25 °C (OD values: 1.49 and 1.4, respectively) than on PCs and CCs (*p* < 0.05). At 35 °C, biofilms were produced at comparable levels on SSCs and CCs. Both strains produced the highest amount of biofilm on SSCs at 25 °C. Figure 1B shows the effect of incubation duration on biofilm formation. Biofilm production by M129-1 and M132-2 was the highest (OD values: 1.49 and 1.4, respectively) on SSCs from the first day (*p* < 0.05) onward, with no significant difference being observed until the sixth day. Biofilm formation by both strains was the lowest on PCs, regardless of the incubation duration (*p* < 0.05).

### 3.4. Survival of Antagonistic LAB Biofilms under Desiccated Conditions

Figure 2 depicts the survival of *L. sakei* M129-1 and *P. pentosaceus* M132-2 (as biofilms formed on SSCs) when exposed to 43% RH at 25 °C for 7 days; *Leuconostoc lactis* M111, which had a weak biofilm-forming ability, was included as a negative control. The population of all three strains decreased significantly over time, regardless of biofilm formation (*p* < 0.05). However, the survival rate of M129-1 and M132-2, which have strong biofilm-forming abilities, was superior (4.7 log CFU/cm^2^ and 5.6 log CFU/cm^2^, respectively) to that of M111 (cultivable cells not detected) following incubation for 5 days.

### 3.5. Antimicrobial Activity of LAB Biofilms on SSCs

Figure 3 depicts changes in populations of five foodborne pathogens, *E. coli, S. aureus, L. monocytogenes, B. cereus,* and *Salmonella*, inoculated on the surface of SSCs with or without LAB biofilms (*L. sakei* M129-1 or *P. pentosaceus* M 132-2) during incubation at 25 °C and 43% RH for 0, 6, 12, 24, or 48 h. The pathogens inoculated on SSCs without biofilm (control) had a decrease of 1.2–3.1 log CFU/cm^2^ within 48 h. In contrast, the number of initial foodborne pathogens inoculated on SSCs with LAB biofilms started to significantly (*p* < 0.05) decrease 6 h after inoculation and decreased by an average of 7.0 log CFU/cm^2^ within 48 h. This indicates that the antibacterial activity of the two LAB isolates was sustained in biofilms.

## 4. Discussion

In the current study, we focused on biofilm formation as well as the antibacterial activity of LAB for food industry applications.

Of the 51 antagonistic microorganisms analyzed, six LAB isolates exhibiting broad antibacterial activity against five foodborne pathogens were isolated from various fermented foods. Among these, two isolates exhibiting the highest levels of antibacterial activity and biofilm-forming ability were selected for further analyses. The selected isolates were identified as *L. sakei* (strain M129-1) and *P. pentosaceus* (strain M132-2). The two isolates have been widely studied as antagonistic microorganisms and reported as biofilm-producing strains [21,22].

Biofilms based on LAB for use in the food industry must possess several properties to compete with pathogenic and spoilage bacteria. These properties include antibacterial activity, adhesion, and desiccation resistance. *L. sakei* M129-1 and *P. pentosaceus* M132-2 isolated in this study showed broad-spectrum antibacterial activity against five foodborne pathogens. Moreover, both isolates exhibited biofilm-forming abilities and desiccation resistance, making them suitable for usage in food production and processing environments.

The interaction between the abiotic surface and the bacterial cell surface for biofilm formation depends on several factors related to cell adhesion, such as hydrophobicity, auto-aggregation, and surface roughness [18]. The initial stage of surface adhesion is controlled by weak hydrophobic interactions between the bacteria and the surface [23]. These interactions become stronger when the microbes begin to synthesize sticky EPSs. In this study, two strains with high hydrophobicity, *L. sakei* M129-1 and *P. pentosaceus* M132-2, showed a strong ability to produce biofilms, indicating a close relationship between the two processes. In addition, high auto-aggregation values of probiotic strains appear to have a positive effect on surface adhesion capacity. Nevertheless, the M113-5 strain showed poor biofilm formation in the crystal violet assay. Among the strains studied, M113-5 was the most hydrophobic and exhibited median auto-aggregation values (Table 2). Proving a clear relationship between cell surface properties and biofilm formation is often difficult because the level of microbial biofilm formation can vary depending upon the culture conditions [18]. Auto-aggregation capacity and bacterial surface hydrophobicity may not be extremely important to biofilm formation, but they are among the special features to keep in mind when considering the complex mechanisms that allow microorganisms to interact with their hosts and exert beneficial effects [24].

Polystyrene surfaces, which are naturally hydrophobic, promote adhesion of microorganisms as compared with surfaces that are hydrophilic, such as those of stainless steel or ceramic [25,26]. Merino et al. [18] observed higher production of Salmonella biofilm on polystyrene than on stainless steel or glass. However, in this study, slightly higher levels of biofilm formation were observed on SSC than on PC or CC (*p* < 0.05). Nevertheless, our results suggest that LAB are able to adhere to, as well as form, biofilms on each of the surfaces tested, regardless of their surface properties. Biofilm formation by the LAB isolates differed significantly in terms of temperature as well as incubation duration. For all surface types, the biofilm levels produced at 15, 25, or 35 °C were significantly higher than those at 5 °C (*p* < 0.05). Further, LAB biofilm formation was promoted significantly at these temperatures when cultured for 3 days on SSC (*p* < 0.05). Although it has been reported that the temperature prevalent during growth affects the adhesion ability by controlling the hydrophobicity of microorganisms [27], the results obtained in this study are attributed to cell growth inhibition at low temperatures. This is evident by the fact that the populations of M129-1 and M132-2 at 5 °C were less than half of those at 25 °C (approximately 4 log CFU/mL, data not shown).

Desiccation, in combination with other strategies, is frequently used in the food processing industry as a means to prevent foodborne pathogen growth [28]. In this study, it was demonstrated that LAB strains M129-1 and M132-2 showed significant resistance to desiccation (atmospheric conditions: 25 °C and 43% RH) in biofilms formed on SSCs (*p* < 0.05). One of the survival strategies of bacteria to overcome environmental stresses, such as desiccation or osmotic pressure, is biofilm formation, whereby cells in a biofilm are encased in a self-producing matrix of EPS that mainly consists of nucleic acids, enzymes, proteins, and lipids, all of which contribute to the cohesion of biofilms [29,30]. Improved desiccation resistance in microorganisms as a result of biofilm formation on abiotic surfaces has been reported by several researchers. Kim et al. [19] compared the resistance of *Pseudomonas extremorientalis*, *Paenibacillus peoriae*, and *Streptomyces cirratus* cells in the form of biofilms on stainless steel surfaces in a dry environment (25 °C, 43% RH) and observed that the resistance of biofilm cells was significantly higher than that of adherent cells (*p* ≤ 0.05; ANOVA with Fisher’s LSD test).

Finally, the ability of the two LAB biofilms to inhibit the growth of foodborne pathogens was examined. SSCs attached to antagonistic biofilms led to the reduction of more than six logs in the counts of the five foodborne pathogens inoculated on these as compared with the controls (SSCs without biofilms) within 12–48 h (*p* < 0.05). This indicates that the antimicrobial activities of the two isolates were retained in the biofilms formed on the SSCs. Considering that the antimicrobial activities of these isolates were undetectable upon neutralization of their culture supernatants (data not shown), these results clearly demonstrate the role of acidic pH in the antagonistic activity of *Lactobacillus*. In general, the production of pH-lowering lactic acids is the most important factor affecting the strength of antimicrobial activity; this includes the production of phenyl lactic acid that exhibits broad antimicrobial activity [31]. Recently, some researchers have reported that LAB biofilms can be used to inhibit foodborne pathogens on the surfaces of food production and processing facilities. In the production of traditional stretched cheeses, non-starter LAB biofilms present in wooden vats inhibited the growth of *L. monocytogenes* and *Salmonella spp*. [32]. Jara et al. [33] reported that the spatial distribution of *L. monocytogenes* Scott A cells was disrupted when the surface was conditioned with *Lactobacillus* biofilm and that these interactions affected the ultimate biofilm structure formed. In addition, Cruciata et al. [34] showed that, in traditional cheese production, barrel-generated non-starter LAB biofilms, owing to their acidity and bacteriocin-producing ability, represent an efficient barrier to the attachment of main dairy pathogens. However, these studies did not analyze the resistance to stress that may occur in the food processing environment, the biofilm formation ability according to the surface material, or the antibacterial activity against various pathogens.

## 5. Conclusions

In the present study, an antagonistic LAB biofilm was developed for application to the surface of food and food processing equipment to minimize the adverse effects of foodborne pathogens. *L. sakei* M129-1 and *P. pentosaceus* M132-2, isolated from fermented Korean food materials, depicted a strong biofilm-forming ability along with hydrophobicity and auto-aggregation. Further, their biofilms inhibited more than six logs of pathogenic bacteria (*B. cereus, L. monocytogenes, Salmonella, S. aureus,* and *E. coli* O157:H7) artificially inoculated on stainless steel surfaces within 12–48 h in a dry environment. The discovery of the formation of such beneficial biofilms might potentially lead to novel biotechnological applications in the food industry.

However, future studies should be conducted on either the same isolate or on other isolates with various characteristics of the LAB strains. In addition, the possibility of LAB cells causing corrosion on abiotic surfaces should be investigated. Finally, LAB effects on pathogenic bacteria present on food contact surfaces must be evaluated on an industrial scale.

## Figures and Tables

**Figure 1 foods-11-00831-f001:**
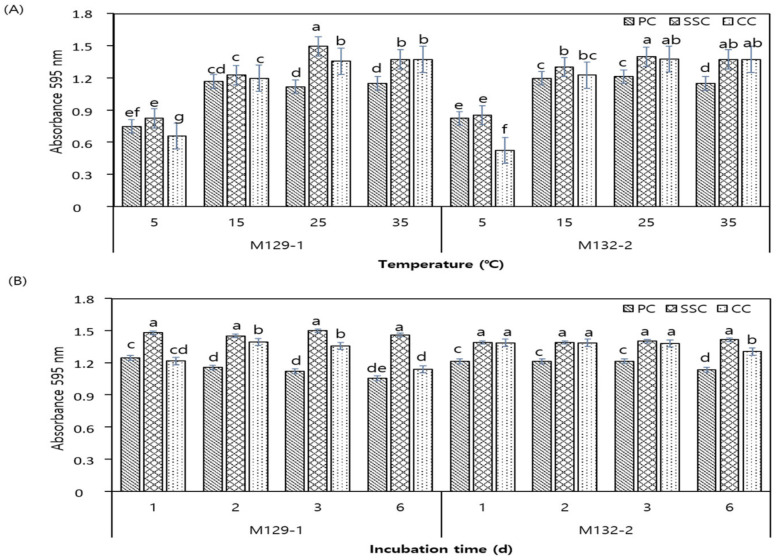
Biofilm formation by *Lactobacillus sakei* M129-1 and *Pediococcus pentosaceus* M132-2 on polystyrene chip (PC), stainless steel chip (SSC), and ceramic chip (CC) at (**A**) 5, 15, 25, or 35 °C for 3 days; and (**B**) 25 °C for 1, 2, 3, or 6 days. Different letters mean a significant difference *(p* < 0.05) between samples, following the Ducan’s multiple test.

**Figure 2 foods-11-00831-f002:**
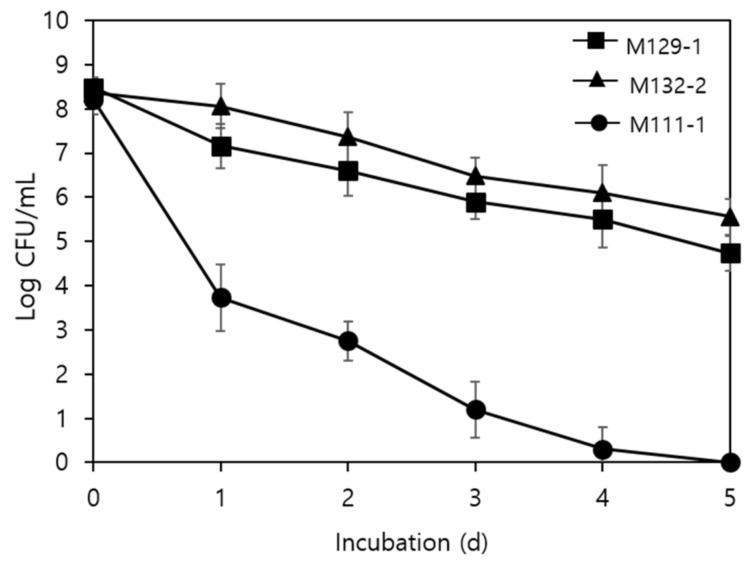
Survival of *Lactobacillus sakei* M129-1, *Pediococcus pentosaceus* M132-2, and *Leuconostoc lactis* M111-1 attached to SSCs under a desiccation environment (43% relative humidity, RH; 25 °C). SSCs were incubated for up to 7 days.

**Figure 3 foods-11-00831-f003:**
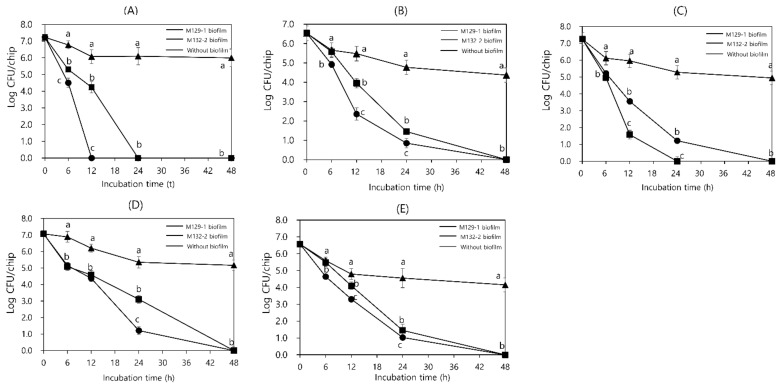
Changes in populations of five pathogens, (**A**) *Escherichia coli* O157:H7, (**B**) *Staphylococcus aureus*, (**C**) *Listeria monocytogenes*, (**D**) *Bacillus cereus*, (**E**) *Salmonella enterica*, with or without biofilms of *Lactobacillus sakei* M129-1 or *Pediococcus pentosaceus* M132-2, when incubated at 25 °C and 43% (RH) for up to 48 h. Values represent the average of samples from each of the three experiments. Error bars represent standard deviations. Different letters mean a significant difference (*p* < 0.05) between samples, following the Ducan’s multiple test.

**Table 1 foods-11-00831-t001:** Inhibitory spectrum of LAB isolates from fermented foods.

	*Escherichia* *coli* O157:H7	*Staphylococcus aureus*	*Salmonella* *enterica*	*Listeria* *monocytogenes*	*Bacillus* *cereus*
M1-5	12.2 ± 0.3	0 ± 0.0	0 ± 0.0	0 ± 0.0	16.6 ± 0.4
M2-8	15.3 ± 0.5	17.9 ± 0.3	11.2 ± 0.5	15.4 ± 0.1	18.1 ± 0.2
M7-2	0 ± 0.0	0 ± 0.0	0 ± 0.0	0 ± 0.0	15.5 ± 0.6
M56	0 ± 0.0	13.5 ± 0.2	0 ± 0.0	0 ± 0.0	0 ± 0.0
M57-1	0 ± 0.0	0 ± 0.0	0 ± 0.0	12.0 ± 0.6	11.0 ± 0.8
M102-3	24.5 ± 0.2	13.6 ± 0.6	22.9 ± 0.5	16.1 ± 0.4	15.6 ± 0.4
M104-1	0 ± 0.0	23.4 ± 0.1	0 ± 0.0	13.6 ± 0.7	0 ± 0.0
M111-1	16.7 ± 0.3	12.2 ± 0.4	17.8 ± 0.5	25.1 ± 0.9	16.2 ± 0.1
M111-3	0 ± 0.0	24.4 ± 0.6	0 ± 0.0	17.3 ± 0.3	0 ± 0.0
M113-5	12.3 ± 0.6	13.7 ± 0.5	15.6 ± 0.2	16.0 ± 0.4	18.6 ± 0.2
M119-3	0 ± 0.0	22.2 ± 0.3	0 ± 0.0	15.2 ± 0.5	25.9 ± 0.6
M123-3	0 ± 0.0	16.5 ± 0.6	0 ± 0.0	11.1 ± 0.7	15.2 ± 0.5
M129-1	22.4 ± 0.5	16.8 ± 0.2	24.0 ± 0.8	25.7 ± 0.2	17.5 ± 0.1
M132-2	20.9 ± 0.6	21.7 ± 0.3	23.2 ± 0.5	25.2 ± 0.2	18.1 ± 0.1
M177-4	0 ± 0.0	0 ± 0.0	0 ± 0.0	16.1 ± 0.4	16.6 ± 0.8
M182-1	0 ± 0.0	0 ± 0.0	0 ± 0.0	23.5 ± 0.7	16.2 ± 0.2
M183-1	0 ± 0.0	0 ± 0.0	0 ± 0.0	15.9 ± 0.5	15.0 ± 0.4
M187-1	0 ± 0.0	0 ± 0.0	0 ± 0.0	17.1 ± 0.2	16.9 ± 0.6
M188-1	0 ± 0.0	0 ± 0.0	0 ± 0.0	12.5 ± 0.6	24.3 ± 0.5
M190-1	0 ± 0.0	0 ± 0.0	0 ± 0.0	16.9 ± 0.4	22.3 ± 0.1
M193-1	0 ± 0.0	0 ± 0.0	0 ± 0.0	17.4 ± 0.9	23.6 ± 0.7
M194-2	0 ± 0.0	0 ± 0.0	0 ± 0.0	17.0 ± 0.6	24.8 ± 0.1
M194-4	0 ± 0.0	0 ± 0.0	0 ± 0.0	18.1 ± 0.2	17.7 ± 0.5
M204-2	0 ± 0.0	0 ± 0.0	0 ± 0.0	25.6 ± 0.1	23.3 ± 0.6
M205-1	0 ± 0.0	0 ± 0.0	0 ± 0.0	16.4 ± 0.2	24.8 ± 0.5
Y101-5	0 ± 0.0	13.2 ± 0.1	0 ± 0.0	11.6 ± 0.6	17.2 ± 0.1
Y103-13	0 ± 0.0	22.0 ± 0.2	0 ± 0.0	16.4 ± 0.2	12.0 ± 0.7
Y105-25	0 ± 0.0	21.4 ± 0.3	0 ± 0.0	16.1 ± 0.3	13.3 ± 0.2
Y105-27	0 ± 0.0	11.5 ± 0.2	0 ± 0.0	11.5 ± 0.9	0 ± 0.0
Y105-3	0 ± 0.0	16.7 ± 0.7	0 ± 0.0	12.8 ± 0.1	0 ± 0.0
Y108-14	0 ± 0.0	0 ± 0.0	0 ± 0.0	16.1 ± 0.5	15.5 ± 0.3
Y108-18	0 ± 0.0	20.9 ± 0.3	0 ± 0.0	15.2 ± 0.6	0 ± 0.0
Y109-10	0 ± 0.0	21.1 ± 0.8	0 ± 0.0	13.9 ± 0.2	0 ± 0.0
Y110-2	0 ± 0.0	0 ± 0.0	0 ± 0.0	13.2 ± 0.1	15.4 ± 0.5
Y110-20	0 ± 0.0	22.4 ± 0.6	0 ± 0.0	12.5 ± 0.3	0 ± 0.0
Y143-5	0 ± 0.0	20.7 ± 0.3	0 ± 0.0	0 ± 0.0	22.4 ± 0.1
Y148-5	0 ± 0.0	0 ± 0.0	0 ± 0.0	12.3 ± 0.1	13.0 ± 0.2
Y149-5	0 ± 0.0	22.5 ± 0.1	0 ± 0.0	15.3 ± 0.3	0 ± 0.0
Y149-6	0 ± 0.0	22.1 ± 0.3	0 ± 0.0	15.8 ± 0.5	0 ± 0.0
Y154-2	0 ± 0.0	0 ± 0.0	0 ± 0.0	16.6 ± 0.2	15.3 ± 0.3
Y154-4	0 ± 0.0	0 ± 0.0	0 ± 0.0	17.4 ± 0.9	17.7 ± 0.5
Y155-2	0 ± 0.0	0 ± 0.0	0 ± 0.0	21.3 ± 0.4	14.8 ± 0.1
Y156-3	0 ± 0.0	0 ± 0.0	0 ± 0.0	20.5 ± 0.6	13.2 ± 0.3
Y205-3	0 ± 0.0	15.5 ± 0.9	0 ± 0.0	12.8 ± 0.6	0 ± 0.0
Y21-2	0 ± 0.0	0 ± 0.0	0 ± 0.0	16.2 ± 0.7	11.3 ± 0.5
Y24-2	0 ± 0.0	0 ± 0.0	0 ± 0.0	11.4 ± 0.9	12.0 ± 0.4
Y28-1	0 ± 0.0	0 ± 0.0	0 ± 0.0	11.7 ± 0.2	12.2 ± 0.5
Y30-1	0 ± 0.0	0 ± 0.0	0 ± 0.0	14.4 ± 0.5	16.8 ± 0.1
Y69-1	0 ± 0.0	0 ± 0.0	0 ± 0.0	15.5 ± 0.4	12.7 ± 0.2
Y69-3	0 ± 0.0	0 ± 0.0	0 ± 0.0	17.1 ± 0.3	13.6 ± 0.9
Y83-2	0 ± 0.0	0 ± 0.0	0 ± 0.0	11.5 ± 0.7	15.1 ± 0.6

Antimicrobial activity was expressed in millimeters as the diameter of the inhibition zone.

**Table 2 foods-11-00831-t002:** Hydrophobicity, auto-aggregation, biofilm formation, and survival rate of each LAB isolate in acidic pH or bile salts.

Isolates	Hydrophobicity (%)	Auto-Aggregation (%)	Biofilm Formation (Crystal Violet Assay)
*Leuconostoc mesenteroides* M2-8	38.40 ± 1.06 ^c,b^	20.53 ± 2.19 ^c^	0.59 ± 0.09 ^c^
*Lactobacillus sakei* M102-3	43.70 ± 2.41 ^b^	22.10 ± 2.56 ^c^	0.89 ± 0.13 ^b^
*Leuconostoc lactis* M111-1	34.76 ± 2.98 ^c^	24.56 ± 3.16 ^c^	0.57 ± 0.10 ^c^
*Lactobacillus curvatus* M113-5	71.21 ± 3.35 ^a^	45.98 ± 3.05 ^b^	0.85 ± 0.08 ^b^
*Lactobacillus sakei* M129-1	69.73 ± 6.11 ^a^	47.53 ± 3.33 ^a,b^	1.48 ± 0.15 ^a^
*Pediococcus pentosaceus* M132-2	68.90 ± 3.53 ^a^	52.28 ± 2.11 ^a^	1.59 ± 0.14 ^a^

Different letters mean a significant difference (*p* < 0.05) between samples, following the Ducan’s multiple test.

## Data Availability

The data presented in this study are available in the article.

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
