# Peer review of "Development of Desiccation-Tolerant Probiotic Biofilms Inhibitory for Growth of Foodborne Pathogens on Stainless Steel Surfaces"

_foods, 2022, doi:10.3390/foods11060831_

Round 1

Reviewer 1 Report

This is a very novel idea. The authors used local benign isolate of lactic acid bacteria to form biofilms in hope of reducing the proliferation and biofilm formation of pathogenic microorganisms. I think the study will be better after undergoing certain revisions in the methodology.

-This study might not be repeatable since the lactic acid bacteria used for the study were simply isolated from the local market and specifications of the strains are not provided. It is also unclear how many strain(s) were selected using this method and if different replications of the study were conducted using the same isolate or several isolates with various characteristics. Some pieces of information are provided in the result section but it would be critical to discuss the number of isolates and their sources in section 2.1. to ensure the study is repeatable in the future if needed.

-Section 2.2. It is unclear how the parameters for the experiment were selected. For example, there is no reference cited to justify the incubation at 37 °C for only 18 hours.

-Section 2.6 is discussing conducting biofilm assay in microplate using crystal violet. The section is clear and proper reference is provided. Section 2.7 however, talks about biofilm formation on abiotic surfaces such as stainless steel and ceramic but does not discuss how the biofilm mass is enumerated. This is usually achieved by glass beads or using sonication methods. The methodology used in a microplate is not appliable to the abiotic surfaces used in the study. Also, there are many methodologies to examine the biofilm formation of bacteria, some are more labor-intensive in the lab but are more accurate. It is critical to discuss the methodologies used and as a limitation of the study discuss other available biofilm formation methodologies that might be more accurate.

-Figure 3 is not statistically analyzed, results of Dunnett’s based ANVOA that authors have used could be incorporated in the graph to ensure the presented material discusses inferential statistics, not just descriptive statistics.

Reviewer 2 Report

The work presented for evaluation, describing the biofilms of lactic acid bacteria as a system protecting against pathogens transmitted by food, is somewhat controversial, but interesting from the cognitive point of view. However, these are rather basic research and cannot be applied in industry. What does the microbial corrosion of materials on which biophyms were created look like? Certainly, such surfaces are subject to corrosion.

Below are some more detailed comments:

Line 19, 21 …:  colony-forming units/cm2 – use abbreviation Log CFU/mL

Line 87: Why was the Listeria monocytogenes strain grown at 37 ° C? The optimal growth temperature for this microorganism is 30 ° C. Please explain.

Table 1.  Sal.enteritidis ??? In the method description, Salmonella enterica KCCM 12021 is mentioned as the strain of Salmonella. Of course, it may be Salmonella enterica ser. Enteritidis, but should have been accounted for before that. Pay attention to the spelling.

In table 1, please provide values in mm and standard deviation.

Why were bile tolerance and low pH or antibiotic resistance studied? These factors have no influence on the formation of biofilms by lactic acid bacteria. It seems to me that it can be removed without harming the manuscript.

 Figure 2. in the description of the OY axis, it should be CFU / mL

Line 299-304 What kind of transference does comparing two different strains within srtA2 gene deficiency? Was the srtA2 gene present in the genome of L. sakei strain (strain M129-1)? If not, remove it from the text.

Line 305:307. “Type I  glyceraldehyde-3-phosphate (GMP) dehydrogenase as well as GMP synthase were upreg- ulated in biofilm cultures of P. pentosaceus, mediating cell adhesion and promoting biofilm  formation [20].” – there is no support in the obtained results.

Line309-313: The listed features - bile resistance, low pH, are not technological features. These parameters are among the unfavorable factors in the human digestive tract. The text is inadequate to the literature. Needs to be redrafted.

Line 317-326: Have the genotypic and phenotypic profiles of the antibiotic resistance of the tested strains been compared?

Based on what indications do the authors claim that demonstrated antibiotic resistance is congenital? On what elements are the antibiotic resistance genes located? Please complete the data. If it is impossible, remove the fragment from the discussion because it has no justification in the results and find it adequate.

Conclusion

The conclusion applies to the skeptics with probiotic characteristics? Did the tested strains have the entire cycle of testing the probiotic features described by the WHO carried out? If so, such information should be included in the manuscript, e.g. as additional materials.

Otherwise, the conclusion must be rewritten.

Round 2

Reviewer 2 Report

Just a request to make minor corrections in table 1.

In table 1, in the place where "0" is, enter "0.0 +/- 0.00. Then the entry is mathematically correct.

After the corrections in Table 1 have been made, I accept the manuscript for publication.